# Photoresponsive supramolecular coordination polyelectrolyte as smart anticounterfeiting inks

Zhiqiang Li[1✉], Xiao Liu[1], Guannan Wang[1], Bin Li[1], Hongzhong Chen[2], Huanrong Li[1✉] & Yanli Zhao [2✉]

While photoluminescence printing is a widely applied anticounterfeiting technique, there are still challenges in developing new generation anticounterfeiting materials with high security. Here we report the construction of a photoresponsive supramolecular coordination poly-electrolyte (SCP) through hierarchical self-assembly of lanthanide ion, bis-ligand and diary-lethene unit, driven by metal-ligand coordination and ionic interaction. Owing to the conformation-dependent photochromic fluorescence resonance energy transfer between the lanthanide donor and diarylethene acceptor, the ring-closure/ring-opening isomerization of the diarylethene unit leads to a photoreversible luminescence on/off switch in the SCP. The SCP is then utilized as security ink to print various patterns, through which photoreversible multiple information patterns with visible/invisible transformations are realized by simply alternating the irradiation with UV and visible light. This work demonstrates the possibility of developing a new class of smart anticounterfeiting materials, which could be operated in a noninvasive manner with a higher level of security.

[1] Tianjin Key Laboratory of Chemical Process Safety, School of Chemical Engineering and Technology, Hebei University of Technology, Tianjin, P. R. China.
[2] Division of Chemistry and Biological Chemistry, School of Physical and Mathematical Sciences, Nanyang Technological University, Singapore, Singapore.
✉email: zhiqiangli@hebut.edu.cn; lihuanrong@hebutedu.cn; zhaoyanli@ntu.edu.sg

Counterfeit goods such as currency, microelectronics, software, movie films, pharmaceutics, and clothing in the market not only cause economic loss to customer and copyright owners, but also bring potential risks to the health and lives of consumers[1–3]. Governments and copyright holders are forced to increase their investments in developing anticounterfeiting technologies. The global market size of anticounterfeiting technologies was 51.8 billion USD in 2017, and the global anticounterfeiting packaging market is expected to grow to 208.4 billion USD in 2023[4]. Amongst the anticounterfeiting techniques and signal outputs, photoluminescence printing is the most widely applied one, because it offers advantages such as easy handling, high-throughput, facile design, and tunable optical properties in multiple dimensions[5–7]. For instance, a series of optical materials, including but not limited to upconversion nanoparticles[8,9], organic dyes[10–12], quantum dots[13], metal-organic frameworks[14,15], and perovskites[16,17], are promising candidates as anticounterfeiting taggants. Lanthanide complexes are also widely applied in anticounterfeiting due to their inherent optical properties, including distinguishable spectroscopic fingerprint, large Stokes shift, and long excited lifetime[18–22]. For example, $Eu^{2+}/Eu^{3+}$ are used in Euro banknotes as a luminescence anticounterfeiting label[23].

However, there are still several challenges in developing new generations of anticounterfeiting materials with more covert and reliable features capable of providing higher security level. (1) Quit a large number of luminescent inks are suspended/dissolved in organic solvents, or contain toxic ions, thus limiting their applications in authenticating food and medicine[4,24]. (2) Authentic information recorded in materials with static optical outputs is often visible under ambient condition or the excitation of UV light[25–27]. Thus, stimulus-responsive materials that can respond to external stimuli and alter their optical outputs would be ideal to bring additional security features, making them more difficult to forge[2,28–31]. On the other hand, invasive stimulus approaches (e.g., thermal, chemical, and mechanical means) may not only contaminate or destroy the goods, but also be inconvenient to operate[32–37]. For example, it is unrealistic for untrained consumer to add acid, alkali or other chemicals to the labels by themselves. Heating approach may cause damage to goods. 3) In terms of printing technologies, inkjet printing is the most common printing form today[38]. In many cases, however, the widespread use of inkjet printing fluorescent nanoparticles/nanocrystals requires either complicated assembly and coating procedures to achieve sufficient loading of nanoparticles and long-term stability of the inks, or the modification of preexisting commercial inkjet printers to cope with high viscosity inks or inks containing oversized nanoparticles (such as aerosol jet printers)[39–42].

To address above-discussed issues, herein, we developed a photoresponsive supramolecular coordination polyelectrolyte (SCP) via the electrostatic interactions of an anionic lanthanide coordination polymer with a cationic photochrome (Fig. 1). Reversible on/off switching of the luminescence signal was realized by remotely alternating UV and visible light irradiation, allowing the fabrication of anticounterfeiting tags for multiple-time verifications. The anionic lanthanide coordination polymer was prepared by the coordination between $Eu^{3+}$ and alkyl bridged bis-2,6-pyridinedicarboxylic acid ligand, followed by mixing with a cationic diarylethene derivative to form SCP in pure water. The diarylethene unit with the features of high photoisomerization yield, excellent fatigue resistance, and thermal irreversibility was chosen as a photoswitch[43–47], since the photochromic fluorescence resonance energy transfer (FRET) between $Eu^{3+}$ and diarylethene unit is typically governed by the conformation of diarylethene[48–50]. Thus, the as-prepared SCP exhibits characteristic emission of $Eu^{3+}$, because the emission spectrum of $Eu^{3+}$ does not overlap with the absorption spectrum of open-form diarylethene. The irradiation of SCP with UV light leads to the isomerization of open-form diarylethene to its close-form conformation, whose absorption band perfectly overlaps with the emission band of $Eu^{3+}$. As a result, the luminescence is quenched due to the activation of the photochromic FRET between $Eu^{3+}$ and photocyclized diarylethene. After subsequent visible light irradiation, the close-form diarylethene isomerizes back to its open-form, and the luminescence intensity is totally recovered. According to this unique property, SCP was filled into a commercially available desktop inkjet printer cartridge to print various high-resolution anticounterfeiting marks. Reversible authentic information with visible/invisible transformation was thus achieved by simple light stimuli, making it suitable for high-security anticounterfeiting applications. Hence, the ring-close and ring-open photoisomerization of the diarylethene moiety regulates the FRET process, leading to reversible luminescence on/off switch in SCP capable of multiple information authentication. In this system, both the lanthanide coordination polymer and the diarylethene derivative are water soluble, and thus, water is the only solvent used in preparing the security ink, enabling its usage in a green condition and its good compatibility with commercial printers. In addition, light irradiation offers clear triggers and spatiotemporal control over the anticounterfeiting patterns in a noninvasive manner.

## Results

**Synthesis and characterization.** Bis-2,6-pyridinedicarboxylic acid ligand (**L**) was synthesized by a two-step procedure and comprehensively characterized (Supplementary Methods and Supplementary Figs. 1–7). Luminescence titration revealed that the coordination stoichiometry between 2,6-pyridinedicarboxylic acid (DPA) and $Eu^{3+}$ is 3:1 (Supplementary Fig. 8), which is inconsistent with the previous report[51,52]. Trimeric lanthanide coordination polymer ($Eu^{3+}$-**L**) was prepared by mixing compound **L** and $EuCl_3$ in water with a molar ratio of 1.5:1 and characterized by FTIR spectra (Supplementary Fig. 9) and $^1H$ NMR spectra. Compared to individual **L**, the absorption band at $1724\ cm^{-1}$ assigned to the $C=O$ stretching vibration of DPA underwent a red shift to $1625\ cm^{-1}$ in the FTIR spectrum of $Eu^{3+}$-**L**, implying the successful coordination of DPA with $Eu^{3+}$ ion[53]. In the $^1H$ NMR spectra (Fig. 2a, b), the proton signals assigned to ligand **L** became highly broadening after the coordination with $Eu^{3+}$, further confirming the formation of the coordination polymer. The high coordination number of $Eu^{3+}$-**L** not only benefits to sufficient sensitization of the $Eu^{3+}$ ion based on the antenna effect, but also prevents luminescent quenching caused by the infiltration of water molecule, thus endowing the lanthanide coordination polymer with characteristic emission color and brightness in both aqueous solution and the solid state under UV light excitation (Supplementary Figs. 10, 11)[54]. The luminescence quantum yield of $Eu^{3+}$-**L** aqueous solution was measured to be 23.31%. The imidazolium salt modified open-form diarylethene (OF-**1**) was synthesized through a robust two-step procedure in a yield of 72%, along with full characterizations (Supplementary Figs. 12–21). UV–Vis (Supplementary Figs. 22–26) and $^1H$ NMR spectra (Supplementary Figs. 27, 28) revealed that compound **1** had excellent reversible ring-open/ring-close photoisomerization behavior (Supplementary Notes 1 and 2).

The lanthanide coordination polymer carries three negative net charges (6COO$^-$ + $Eu^{3+}$) per coordination center, allowing it to further assemble with positively charged OF-**1** based on electrostatic interaction[55–59]. The SCP ($Eu^{3+}$-**L**-OF-**1**) was then prepared by mixing $Eu^{3+}$-**L** and OF-**1** at charge stoichiometry ($Eu^{3+}$:OF-**1** = 1:1.5). Zeta potential experiments were carried out to verify the existence of electrostatic interaction between $Eu^{3+}$-**L**

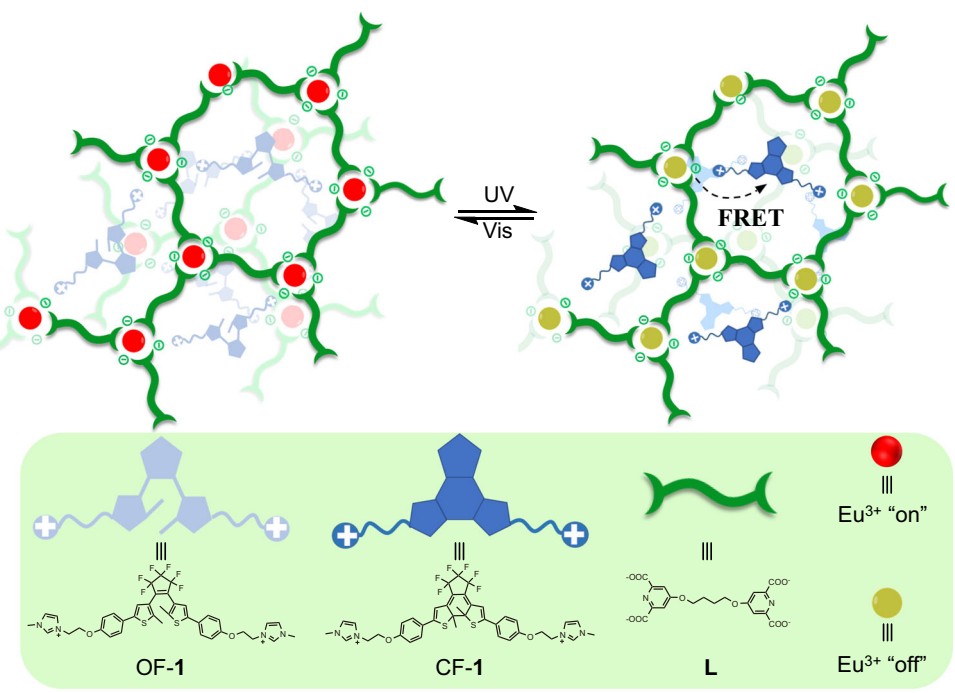

**Fig. 1 Schematic illustration.** The construction of the photochromic supramolecular coordination polyelectrolyte, and the chemical structures of corresponding components.

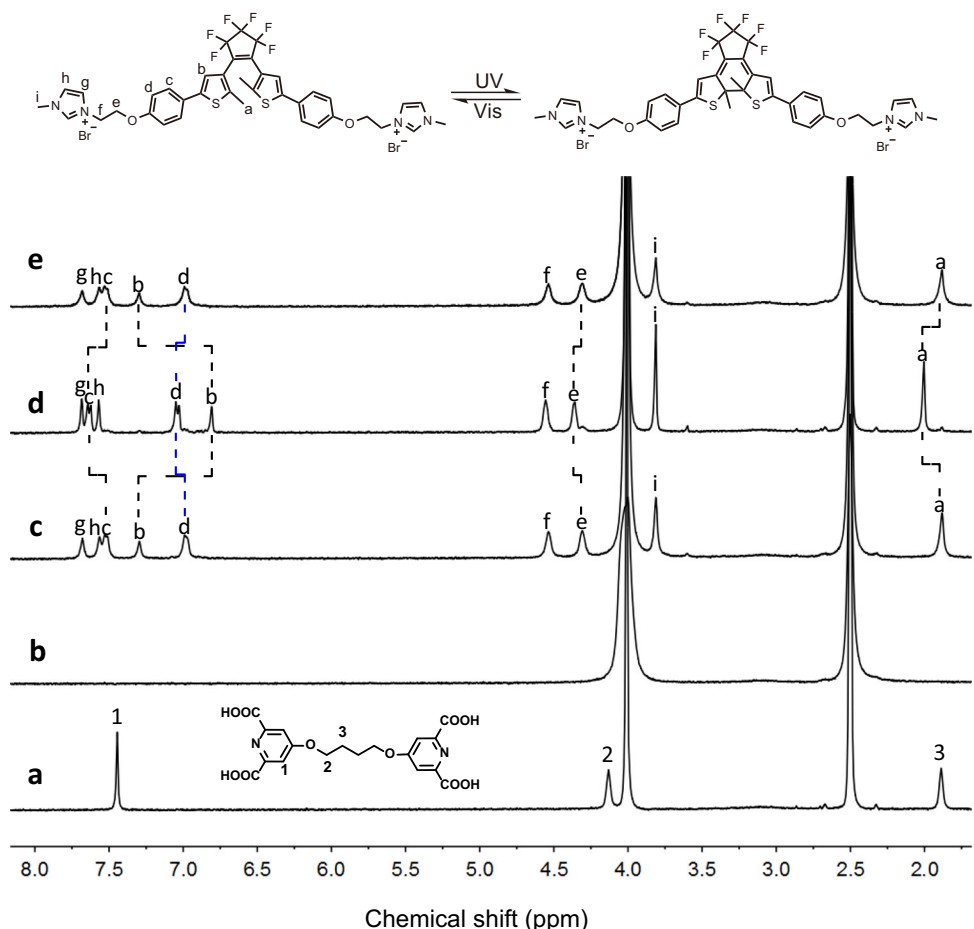

**Fig. 2 ¹H NMR spectral studies.** Partial ¹H NMR spectra (DMSO:$D_2O$ = 4:1, 400 MHz, 25 °C) of **a** compound **L**, **b** $Eu^{3+}$-**L**, **c-e** $Eu^{3+}$-**L**-OF-**1**: **c** before and **d** after the irradiation by UV light (300 nm, 60 min), and **e** subsequent irradiation with visible light (> 450 nm, 60 min). [$Eu^{3+}$] = 1.4 × 10⁻⁴ M, [**L**] = [OF-**1**] = 2.1 × 10⁻⁴ M.

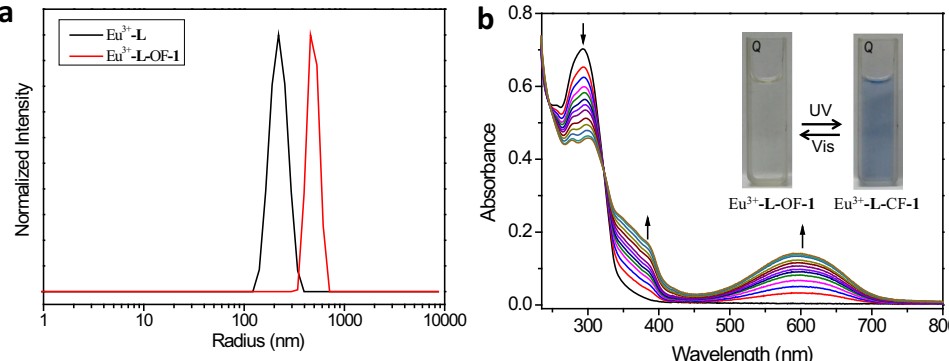

**Fig. 3 DLS size distribution and UV–Vis spectral studies. a** DLS size distribution of Eu$^{3+}$-**L** (black curve) and Eu$^{3+}$-**L**-OF-**1** (red curve) ([Eu$^{3+}$] = 1.4 × 10$^{-4}$ M, [**L**] = [OF-**1**] = 2.1 × 10$^{-4}$ M). **b** UV–Vis spectral changes and corresponding photographic images of Eu$^{3+}$-**L**-OF-**1** and Eu$^{3+}$-**L**-CF-**1** with alternating 300 nm UV and >450 nm visible light irradiation in water for up to 60 s each time ([Eu$^{3+}$] = 1.4 × 10$^{-5}$ M, [**L**] = [OF-**1**] = 2.1 × 10$^{-5}$ M).

and OF-**1** (Supplementary Fig. 29). Individual Eu$^{3+}$-**L** displayed a negative potential of −19.53 mV, while the ζ-potential value of individual OF-**1** was measured to be 20.15 mV. The Eu$^{3+}$-**L**-OF-**1** solution was almost electrically neutral (1.54 mV). These results confirmed the presence of electrostatic interaction between Eu$^{3+}$-**L** and OF-**1**, enabling suitable distance between the energy donor and acceptor. Dynamic light scattering (DLS) measurements confirmed the formation of supramolecular assembly between Eu$^{3+}$-**L** and OF-**1**. The DLS experiment of Eu$^{3+}$-**L** (Fig. 3a) shows a hydrodynamic radius of 220 nm, indirectly proving the formation of large-scaled coordination polymer in solution. The hydrodynamic radius of Eu$^{3+}$-**L**-OF-**1** increases to around 500 nm, much larger than that of Eu$^{3+}$-**L**, revealing that Eu$^{3+}$-**L** assembles with OF-**1** to form the supramolecular polymer[60,61]. Meanwhile, uniform spheres with an average diameter of 300 nm were observed by transmission electron microscopy (Supplementary Fig. 30), providing intuitive evidence for the formation of the self-assembly between Eu$^{3+}$-**L** and OF-**1**.

**Photoresponsive property**. We then investigated the photoresponsive property of Eu$^{3+}$-**L**-OF-**1** resulting from the isomerization of the diarylethene moiety. The UV–Vis spectra of Eu$^{3+}$-**L**-OF-**1** (Fig. 3b) showed an absorption band at 294 nm corresponding to OF-**1** unit, and no absorption over 400 nm was observed. Upon the irradiation with UV light (300 nm), the absorption at 294 nm gradually decreased, two new absorption bands centered at 380 nm and 596 nm appeared. Meanwhile, the colorless aqueous solution changed to dark blue (insert of Fig. 3b). These phenomena jointly demonstrated the OF-**1** transformed to its close form (CF-**1**) after the irradiation. All these changes levelled off in 60 s (Supplementary Fig. 31). Moreover, a well-defined isosbestic point was observed at 323 nm, indicating that ring-open isomer cleanly transformed into the photocyclized form in SCP[62,63]. We further measured the photocyclization yield at the photostationary state by $^1$H NMR spectra (Fig. 2c,d). Since the proton signals of OF-**1** showed serious broadening in aqueous media (Supplementary Fig. 32), the $^1$H NMR spectral study was carried out in mixed deuterated solvent (DMSO-d$_6$:D$_2$O = 4:1). After irradiated by UV light (300 nm, 60 min), the thiophene protons (H$_b$) underwent an obvious upfield shift from 7.30 to 6.81 ppm, mainly due to the electronic shielding effect in the large conjugated closed ring isomers[64]. The methyl protons (H$_a$) of the diarylethene unit underwent an apparent downfield shift from 1.88 to 2.00 ppm. Meanwhile, the aromatic protons H$_c$ and H$_d$ showed downfield shifts from 7.52 ppm to 7.63 ppm and from 6.98 ppm to

7.04 ppm, respectively. All these shifts were thorough, and no apparent residual peaks retained in the original chemical shifts after the UV light irradiation (Supplementary Fig. 33). The molar ratio of CF-**1**: OF-**1** was determined to be 0.94:0.06 according to the integrating resonance of protons H$_a$, indicating nearly quantitative (~94%) conversion from Eu$^{3+}$-**L**-OF-**1** to Eu$^{3+}$-**L**-CF-**1** upon exposure to UV light[65]. Interestingly, a complete recovery in both UV–Vis (Supplementary Figs. 34, 35) and $^1$H NMR spectra (Fig. 2e) was achieved upon subsequent irradiation of the resulting Eu$^{3+}$-**L**-CF-**1** solution with >450 nm visible light, accompanied by color change back to colorless, revealing that this photoisomerization behavior was fully reversible.

The photoresponsive luminescent behavior of SCP was then investigated. In the as-prepared Eu$^{3+}$-**L**-OF-**1**, no FRET was observed, because there was no spectral overlapping between the UV–Vis absorption of OF-**1** and the emission spectrum of Eu$^{3+}$-**L**. Eu$^{3+}$-**L**-OF-**1** exhibited the characteristic spectral line of lanthanide. The excitation spectrum of Eu$^{3+}$-**L**-OF-**1** showed a broad band centered at 265 nm, attributed to the absorption of the DPA moiety (Supplementary Fig. 36). The corresponding emission spectrum was composed of five sharp peaks at 580, 594, 615, 649, and 692 nm, referred to the $^5D_0$ to $^7F_J$ ($J$ = 0-4) transitions of Eu$^{3+}$ respectively, in which the $^5D_0 \rightarrow {}^7F_2$ transition at 615 nm is dominant and responsible for the bright red emitting color (Supplementary Fig. 36)[66]. On the other hand, the luminescence emission spectrum of lanthanide coordination polymer Eu$^{3+}$-**L** completely overlapped with the absorption spectrum of CF-**1** in the range of 500-700 nm (Fig. 4a), implying that efficient FRET process may occur from Eu$^{3+}$ to CF-**1** in Eu$^{3+}$-**L**-CF-**1**. As expected, the luminescence of Eu$^{3+}$ (Fig. 4b) was quenched gradually upon irradiating SCP with UV light. The luminescence quenching followed a biexponential attenuation law, containing a fast process, followed by a slow process to the photostationary state in 60 s (inset of Fig. 4b)[49]. The luminescence intensity was quenched completely at the end, and the decay decreased from 1,289 to 12 μs (Supplementary Figs. 37–41), with concomitant decrease of the luminescence quantum yield from 15.84% to 0.85%. These phenomena confirmed the occurrence of the FRET process with an efficiency ($E$) of 98%, calculated according to the reported method[67]. The quenched luminescence of Eu$^{3+}$-**L**-CF-**1** could completely recover to its original level upon subsequent visible light irradiation, ascribing to the photocycloreversion reaction (Fig. 4c). In particular, the photocontrolled luminescence on/off switch of SCP presented outstanding reversibility, and no apparent deterioration in the luminescence intensity (less than 4%) was observed after 20 consecutive cycles of alternating UV and visible light irradiations (Fig. 4d). Thus, SCP exhibited excellent

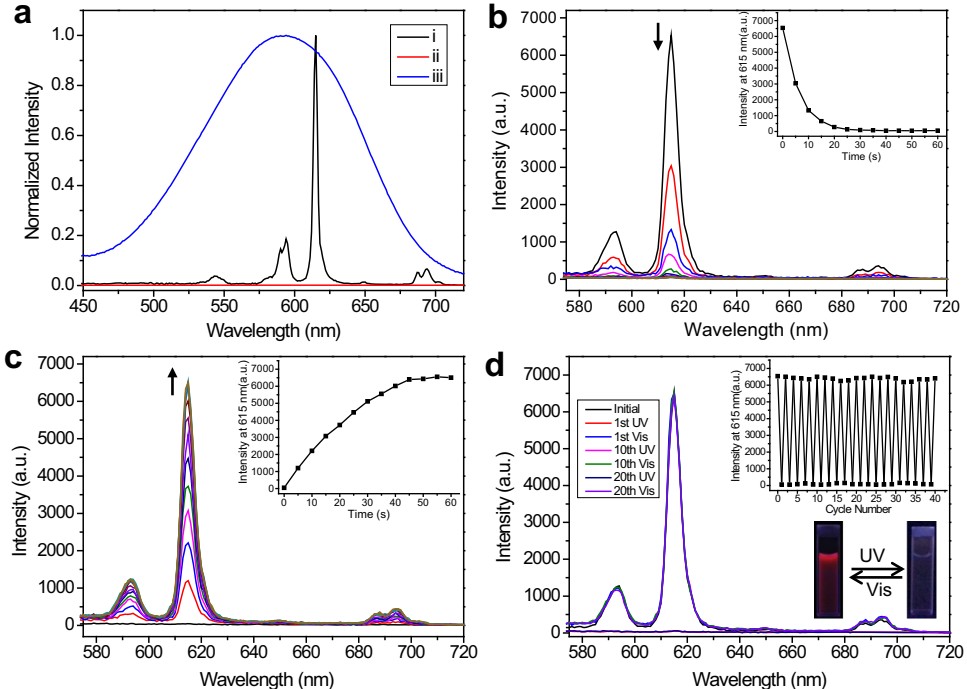

**Fig. 4 Photophysical studies. a** Partial emission spectrum (black curve) of Eu$^{3+}$-**L**, and absorption spectra of OF-**1** (red curve) before and (blue curve) after irradiation with 300 nm UV light for 60 s. **b**, **c** Luminescence emission spectral changes of Eu$^{3+}$-**L**-OF-**1** upon (**b**) UV light (300 nm) irradiation and **c** subsequent visible light (>450 nm) irradiation in water. Insets show corresponding emission intensity changes at 615 nm. **d** Luminescence emission changes of Eu$^{3+}$-**L**-OF-**1** upon consecutive alternating exposure to UV and visible light. Insets show corresponding intensity changes at 615 nm (upper) and the photographs of the SCP solution under 254 nm UV lamp (lower). [Eu$^{3+}$] = 1.4 × 10$^{-4}$ M, [**L**] = [OF-**1**] = 2.1 × 10$^{-4}$ M.

fatigue resistance, which is of utmost importance for multiple anticounterfeiting applications.

It is worth noticing that the diarylethene derivative is bistable[37], which means that the spontaneous photocycloreversion reaction is extremely slow under natural conditions. The half-life ($t_{1/2}$) of Eu$^{3+}$-**L**-CF-**1** at 25 °C was estimated to be 376.7 min (Supplementary Figs. 42, 43, Supplementary Table 1, and Supplementary Note 3), ranking one of the longest $t_{1/2}$ values reported so far in diarylethene derivatives[68,69], which confirmed that the self-switching is negligible. Only slight self-switching of Eu$^{3+}$-**L**-CF-**1** was observed upon continuous exposure to sunlight for 90 min (Supplementary Fig. 44). When the Eu$^{3+}$-**L**-CF-**1** solution was kept at an elevated temperature (60 °C) in the dark, no sign of thermal ring opening was observed from the UV–Vis spectra, supporting the good thermal stability of Eu$^{3+}$-**L**-CF-**1** (Supplementary Fig. 45).

**Pattern printing**. The developed SCP with important features of rapid response, prominent anti-fatigue capability and thermally irreversible luminescence on/off photoswitch encouraged us to further explore its performance in smart anticounterfeiting. We directly filled the Eu$^{3+}$-**L**-OF-**1** aqueous solution in a commercial inkjet printer (canon PIXMA ip1180) cartridge with the concentration low to 2.1 × 10$^{-4}$ M (according to the concentration of OF-**1**), and printed various high-resolution quick response (QR) codes on commercial blue polyester terephthalate (PET) films (Fig. 5a, b). The obtained QR code was invisible under daylight due to the colorless nature of Eu$^{3+}$-**L**-OF-**1** aqueous solution (Fig. 5c and Supplementary Movie 1). However, bright red luminescent pattern was observed under 254 nm UV lamp, allowing to retrieve the encoded information quickly and accurately by scanning through a smartphone (Fig. 5d and Supplementary Movie 2). It should be noted that the UV absorbance

intensity of OF-**1** at 254 nm is low, and thus the conversion from OF-**1** to CF-**1** under 254 nm UV lamp is very slow, providing enough time for recognizing the authentic information recorded in the QR code. The luminescence was quenched upon 300 nm UV light irradiation, making the QR code invisible under UV light. Although the pattern turned to blue under daylight, it can be completely masked by the blue background of the PET film. Through which, the absolutely and really invisible security pattern was achieved under both daylight (Fig. 5e and Supplementary Movie 3) and UV light (Fig. 5f and Supplementary Movie 4), which was highly sufficient for confidential information encryption.

As discussed above, CF-**1** is the photostable state under daylight, especially in solid state. Consequently, the erased pattern remained unreadable even after placing under sunlight for one month (Supplementary Movies 5 and 6). The erased pattern could be completely recovered and recognized upon further irradiating with visible light (>450 nm). Moreover, even after 20 consecutive switching cycles, the quality of the remote light triggered information pattern with visible/invisible transformation process still remained unaffected (Supplementary Movies 7 and 8). Thus, the rapid response, noninvasive regulation, excellent fatigue resistance, and thermal irreversibility of SCP-based system made it a suitable anticounterfeiting ink for multiple authentic information encryption and decryption.

## Discussion
In summary, we have developed a hierarchical self-assembly approach to realize a photoresponsive supramolecular coordination polyelectrolyte capable of reversible multiple information encryption and decryption. An anionic lanthanide coordination polymer, obtained from the coordination between lanthanide ion and a bis-ligand, further assembles with a cationic diarylethene

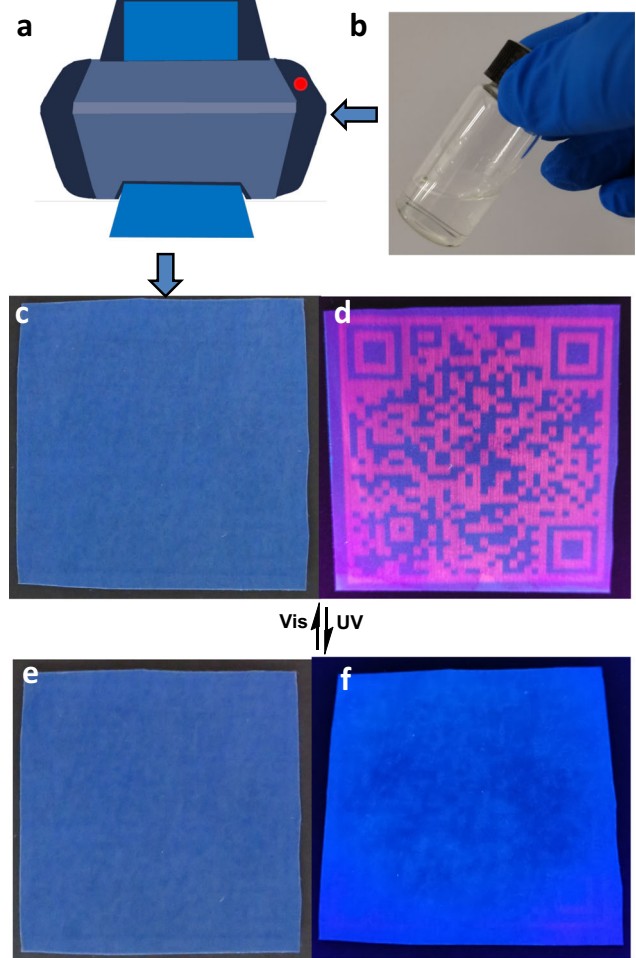

**Fig. 5 Pattern printing using SCP as the ink. a, b** Schematic illustration of the pattern printing process. Light triggered QR code with visible/invisible transformation behavior was achieved by using supramolecular coordination polyelectrolyte (SCP) as the smart ink. **c–f** Digital photos of SCP-based luminescent QR code on commercial blue PET film (size: 5 × 5 cm) upon alternating UV (300 nm, 60 s) and visible light (>450 nm, 120 s) irradiation. **c, d** Photos under daylight. **e, f** Photos under 254 nm UV lamp. [Eu$^{3+}$] = 1.4 × 10$^{-4}$ M, [**L**] = [OF-**1**] = 2.1 × 10$^{-4}$ M in the SCP ink.

derivative based on ionic interaction to afford the SCP. Significantly, the ring-open/ring-close photoisomerization of the diarylethene moiety governs the FRET process between the lanthanide emitting center and the diarylethene component, leading to reversible luminescence on/off switch in the SCP. This SCP has been directly utilized as a security ink to realize reversible authentic information patterning with visible/invisible transformation by simply alternating the exposure to UV and visible light. The developed materials and its associated patterning technology with environmentally friendly preparation process, remote light control, rapid response, excellent fatigue resistance and thermal irreversibility have demonstrated a promising potential as a high-security anticounterfeiting ink in various fields, including authenticating food and medicine.

## Methods

Synthetic routes for compounds **L** and OF-**1** are shown in Supplementary Figs. 1, 12.

**Synthesis of compound 5**. A mixture of compound **6** (474 mg, 1.98 mmol), 1,4-dibromobutane (60 μL, 0.5 mmol) and K$_2$CO$_3$ (248 mg, 1.8 mmol) was stirred in *N,N*-dimethylformamide (10 mL) under N$_2$ at 80 °C for 48 h. Then, the reaction mixture was poured into water (100 mL). The resulting white precipitate was

collected and washed three times with water. The precipitate was then dissolved in CH$_2$Cl$_2$ (100 mL) and washed with a solution of 5 % aqueous NaOH solution (2 × 50 mL). The organic phase was concentrated and dried under vacuum to give compound **5** as a white solid in 80% yield. $^1$H NMR (400 MHz, CDCl$_3$, ppm): δ 7.79 (s, 4H), 4.48 (q, *J* = 7.1 Hz, 8H), 4.27 (m, 4H), 2.08 (m, 4H), 1.46 (t, *J* = 7.1 Hz, 12H). $^{13}$C NMR (100 MHz, CDCl$_3$, ppm): δ 166.7, 164.7, 150.2, 114.1, 68.2, 62.4, 25.3, 14.2. HRMS [M + H]$^+$ calcd. for C$_{26}$H$_{33}$N$_2$O$_{10}^+$ 533.2135; found: 533.2126; Anal. Cald. for C$_{26}$H$_{32}$N$_2$O$_{10}$: C, 58.64; H, 6.06; N, 5.26; Found: C, 58.58; H, 6.10; N, 5.22.

**Synthesis of compound L**. A mixture of compound **5** (168 mg, 0.4 mmol), KOH (224 mg, 4 mmol), methanol (10 mL) and water (10 mL) was stirred at 60 °C for 12 h, and then acidified with HCl (3 M) to pH 4. The precipitate was collected by centrifugation, washed with H$_2$O, and dried under vacuum to give compound **L** as a white solid in 60% yield. $^1$H NMR (400 MHz, D$_2$O, ppm): δ 7.59 (s, 4H), 4.34 (m, 4H), 2.08 (m, 4H). $^{13}$C NMR (100 Hz, H$_2$O, ppm): δ 172.7, 166.7, 154.9, 111.4, 68.3, 24.7. HRMS [M-H]$^-$ calcd. for C$_{18}$H$_{15}$N$_2$O$_{10}^-$ 419.0727; found: 419.0734; Anal. Cald. for C$_{18}$H$_{16}$N$_2$O$_{10}$: C, 51.44; H, 3.84; N, 6.66; Found: C, 51.29; H, 3.78; N, 6.60.

**Preparation of the coordination polymer Eu$^{3+}$-L**. Compound **L** (42 mg, 0.1 mmol) and potassium hydroxide (22.4 mg, 0.4 mmol) were dissolved in water (10 mL). Then, europium chloride hexahydrate (24.5 mg, 0.067 mmol) was added with stirring for 30 min. The mixture was dried under vacuum to give Eu$^{3+}$-**L** as a white solid.

**Synthesis of compound 4**. 4-Hydroxyphenylboronicacidpinacolester (220 mg, 1 mmol), 1,2-dibromoethane (940 mg, 5 mmol), and K$_2$CO$_3$ (690 mg, 5 mmol) were added into acetonitrile (20 mL) with stirring. The mixture was heated at 70 °C under N$_2$ atmosphere for 24 h. After cooling down to room temperature, the reaction mixture was filtered and the residue was washed with CH$_2$Cl$_2$. Then, the filtrate was concentrated under reduced pressure. The residue was dissolved by CH$_2$Cl$_2$ (50 mL) and washed twice with saturated NaCl solution. The organic phase was concentrated. The crude product was purified by column chromatography over silica gel (eluent: petroleum ether/ethyl acetate = 20:1), and compound **4** was obtained as white powder in 80% yield. $^1$H NMR (400 MHz, CDCl$_3$, ppm): δ 7.75 (d, *J* = 8.5 Hz, 2H), 6.90 (d, *J* = 8.5 Hz, 2H), 4.32 (t, *J* = 6.4 Hz, 2H), 3.64 (t, *J* = 6.3 Hz, 2H), 1.33 (s, 12H). $^{13}$C NMR (100 MHz, CDCl$_3$, ppm): δ 160.6, 136.6, 114.0, 83.6, 67.6, 28.9, 24.9. HRMS [M + H]$^+$ calcd. for C$_{14}$H$_{21}$BBrO$_3^+$ 327.0767; found: 327.0754; Anal. Cald. for C$_{14}$H$_{20}$BBrO$_3$: C, 51.42; H, 6.16; Found: C, 51.38; H, 6.18.

**Synthesis of compound 2**. Compound **4** (327 mg, 1 mmol), compound **3** (210 mg, 0.4 mmol), Pd(PPh$_3$)$_4$ (70 mg, 0.06 mmol), and Na$_2$CO$_3$ (680 mg, 6.4 mmol) were added to a mixed solution of water (4 mL) and dimethoxyethane (30 mL). The mixture was refluxed under N$_2$ at 90 °C in dark for 24 h. After cooling down to room temperature, the solvent was removed under vacuum. The residue was extracted by dichloromethane, and purified on a silica gel column using petroleum ether/ethyl acetate (20:1) as the eluent. $^1$H NMR (400 MHz, CDCl$_3$, ppm): δ 7.47 (d, *J* = 8.6 Hz, 4H), 7.17 (s, 2H), 6.93 (d, *J* = 8.6 Hz, 4H), 4.32 (t, *J* = 6.2 Hz, 4H), 3.66 (t, *J* = 6.2 Hz, 4H), 1.94 (s, 6H). $^{13}$C NMR (100 MHz, CDCl$_3$, ppm): δ 158.0, 141.9, 140.5, 127.0, 126.9, 125.8, 121.5, 115.2, 68.0, 28.9, 14.5. HRMS [M + H]$^+$ calcd. for C$_{31}$H$_{25}$Br$_2$F$_6$O$_2$S$_2^+$ 766.9546; found: 766.9537; Anal. Cald. for C$_{31}$H$_{24}$Br$_2$F$_6$O$_2$S$_2$: C, 48.58; H, 3.16; Found: C, 48.51; H, 3.19.

**Synthesis of compound OF-1**. Compound **2** (383 mg, 0.5 mmol) was dissolved in acetonitrile (10 mL), and then 1-methylimidazole (410 mg, 5 mmol) was added. The reaction mixture was stirred at 80 °C for 12 h. After cooling down to room temperature, the obtained precipitate was collected by centrifugation and washed with diethyl ether for three times to afford the desired product OF-**1** in 90% yield. $^1$H NMR (400 MHz, DMSO-*d$_6$*, ppm): δ 9.20 (s, 2H), 7.83 (s, 2H), 7.73 (s, 2H), 7.58 (d, *J* = 8.4 Hz, 4H), 7.39 (s, 2H), 7.02 (d, *J* = 8.6 Hz, 4H), 4.61 (t, *J* = 4.6 Hz, 4H), 4.39 (t, *J* = 4.7 Hz, 4H), 3.88 (s, 6H), 1.93 (s, 6H). $^{13}$C NMR (100 MHz, DMSO-*d$_6$*, ppm): δ 158.1, 142.0, 140.7, 137.5, 127.2, 126.4, 125.4, 124.0, 123.3, 121.8, 115.8, 66.4, 48.8, 36.3, 14.5. HRMS [M-2Br]$^{2+}$ calcd. for C$_{39}$H$_{36}$Br$_2$F$_6$N$_4$O$_2$S$_2^{2+}$ 385.1086; found: 385.1081. Anal. Cald. for C$_{39}$H$_{36}$Br$_2$F$_6$N$_4$O$_2$S$_2$: C, 50.33; H, 3.90; N, 6.02; Found: C, 50.39; H, 3.98; N, 5.94.

**Preparation of the QR code**. In a standard procedure, commercial PET film was first printed with blue background, and the QR-pattern was then directly printed on the blue PET film. The QR code was scanned by a commercially available smartphone APP.

## Data availability

The authors declare that the data supporting the findings of this study are available within the article and its Supplementary Information. Extra data are available from the corresponding authors upon reasonable request.

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

## Acknowledgements

This work was financially supported by the National Natural Science Foundation of China (21871075, 21502039, and 21771050), the Natural Science Foundation of Hebei Province (B2018202134, B2016202149, B2016202147, and B2017202048), the Tianjin Natural Science Foundation (19JCQNJC04900), the Program for Top 100 Innovative Talents in Colleges and Universities of Hebei Province (SLRC2019023), the Singapore Academic Research Fund (RT12/19), and the Singapore Agency for Science, Technology, and Research (A*STAR) AME IRG grant (A1883c0005).

## Author contributions

Z.L. designed the experiments and drafted the manuscript. Z.L., X.L., G.W., B.L., and H.C. performed the experiments and analyzed the data. H.L. and Y.Z. supervised the work and edited the manuscript.

## Competing interests

The authors declare no competing interests.
