## [Peer Review File · Nature Communications]

Reviewers' Comments:

Reviewer #1:

Remarks to the Author:

This is an interesting paper describing the preparation of photoswitchable supramolecular coordination polyelectrolyte as a smart anticounterfeiting ink. The authors regulate the FRET process between the lanthanide emitting center and the diarylethene component by light irradiation, and the resulting smart ink can be readily prepared in pure water and fully compatible with commercial printers. The features of environmentally friendly preparation process, remote light control, rapid response, and good fatigue resistance demonstrate a promising system as an anticounterfeiting ink in various fields. Thus, I recommend the publication of the work after addressing the following minor comments.

(1) The authors assigned the absorption band at 294 nm in the UV-Vis spectrum of Eu^{3+} -L-OF-1 to the diarylethene unit. However, the absorption of 2,6-pyridinedicarboxylic acid is also near. Is there any overlap between their absorbance? In this context, UV-Vis spectrum of Eu^{3+} -L should be provided for comparison.

(2) The authors measured the luminescence decays before and after UV light irradiation. I would suggest that they also measure and compare the luminescence quantum efficiency.

(3) The luminescence of the ink (Figure 3D inset) is not as bright as that in Figure S10, perhaps due to different lanthanide concentrations used. Nevertheless, the comparison of luminescence intensity between individual lanthanide complex and the ink should be provided.

(4) In the elemental analysis, the authors only provide the C/H/N data of L and OF-1. Elemental analysis results of major intermediates should also be given.

(5) The authors claimed that "the proton signals assigned to L became highly broadening after the coordination with Eu^{3+} ", and indeed no obvious NMR peak was observed in Figure 1B. Have the authors tried to carry out this experiment at higher concentrations?

Reviewer #2:

Remarks to the Author:

Stimuli-responsive materials are important for smart anticounterfeiting. In this work, the authors report a photoresponsive anticounterfeiting ink composed of lanthanide complex and diarylethene unit in pure water. The mechanism behind was investigated and explained convincingly. The work is not only accessible, but also of interest and relevance to a broad readership. Compared to the reported stimuli-responsive materials dissolved/suspended in organic solvents, direct printing in pure water with commercial inkjet printer is especially appealing. The concept is full of attraction, and I believe that this is a breakthrough research in the field. Therefore, I recommend the publication of this interesting manuscript in Nature Communications after minor revisions as stated below:

1. What is the major function of supramolecular coordination polyelectrolyte used? Could the same effect be easily accomplished by mixing the DAE molecule with Eu complex through simple solution processing? The relevant control experiments should be performed.
2. In addition to the Eu complex, can other lanthanide metal complexes or organic dyes with red emission be used for the same purpose?
3. The absorption changes in the UV-Vis spectra upon UV and visible light irradiation, and the luminescence quenching curves showed a fast process, followed by a slow process. However, the luminescence recovery curve in Figure 3C showed a slow process in the early stage of visible light illumination. What reasons cause this phenomenon, and would this phenomenon affect the rapid response in smart anti-counterfeiting?

4. In Figure 4, the authors first irradiated the pattern with 300 nm light, and then irradiated it with > 450 nm light. What if the pattern is directly irradiated with > 450 nm, and will it directly show luminescence just like when irradiated with UV light?
5. For better clarity of the data, more specific explanations should be added to the caption of Figure 4. In Figure 4C and 4E, the background is dark blue, while in Figure 4D and 4F, it is light blue. Please make them consistent.
6. The FRET efficiency (E) can be calculated according to either the luminescence intensity or the luminescence lifetime data. Which method did the authors use? Is there any difference about the E values obtained from the two methods?

Reviewer #3:

Remarks to the Author:

The paper by Zhao and Li et al reported a coordination supramolecular system displaying reversible photo-responsive emission and absorbance changes which is used for the purpose of anti-counterfeiting. The main problem of this work is lack of novelty since both the concept of anti-counterfeiting use photo-responsive materials and the compound L is quite old. Especially, L is well-studied 17~18 years ago (1. Vermonden T et al, *Macromolecules*, 2003, 36,7035; *Eur. J. Inorg. Chem.*, 2003, 2847; *JACS*, 2004,126, 15802) and the coordination between L and Eu³⁺ was well-studied in a number of refs in recent years (Yang L et al, *Soft Matter*, 2011,7,2720; Xu et al, *Soft Matter*, 2014, 10, 4686; Wang et al, *Chem Commun*, 2013,49,3736; *Macromolecules* 2019, 52, 8643; *Soft Matter*, 2020, 16,2953.). It is a pity that Zhao et al is trying to hide this fact and never cite any of these references. In SI they cite some refs about the synthesis of the materials that is necessary for the synthesis of L, and deliberately citing ref 51 to support their conclusion about the coordination ratio between the DPA ligand. Actually, the coordination between L and Eu was already very clearly discussed in the above refs. The synthesis route of L they adopted is exactly the same as Vermonden et al. So, it is a pity that the authors never mention any of these works, and trying hard to pretend they are working on a new compound.

Other problems:

1. The coordination between L and Eu would not form polymer at concentrations they use. Only small coordination complex is possible since the coordination is reversible and dynamic. The experimental evidence for the formation of small complex can be found in refs by Vermonden, Yang, and Wang mentioned above;
2. They did not provide direct evidence for the formation of supramolecular assembly between L-Eu and OF-1. Ionic interaction indeed occurs, but this does not mean the formation of supramolecular self-assembly. They should investigate the form of the ionic complexes.
3. Figure 1 and Figure S10, to study formation of Eu-L coordination complex, is not necessary since this coordination chemistry is already well-studied in literature.
4. Using color change as anti-counterfeiting The QY for rare earth metal is usually below 21%. The authors need to discuss the QY for emission-based anti-counterfeiting.

Reviewer #4:

Remarks to the Author:

Z. Li, H. Li, and Y. Zhao report a material system based on bifunctional supramolecular building blocks that incorporate dithienylethylene (DTE) photoswitches and Europium (Eu) ions to act as crosslinkers and emitters. In the assembled material, photoswitching of DTE results in a change in the electronic levels of the DTE making or preventing FRET from the Eu to the DTE. In the closed

state, where FRET is possible, there is no emission from the Eu-ions and the material appears dark. In the open state there is no FRET and the material exhibits red photoluminescence typical for Eu-containing materials. The authors apply the molecular material as inks and produce QR codes using ink-jet printing. After deposition of the ink, the solvent evaporates, driving the material system into the above described supramolecular self-assembly. The authors foresee that a potential application of this material will be for anti-counterfeiting labels. In importance for such anti-counterfeiting systems is explained in the introduction of the paper.

The manuscript at hand represents a sensible collaboration of the two individual groups with their respective expertise in supramolecular assembly and photonic material systems.

While I find the work suitable for nature communications, I have the following questions and remarks, which are not clearly described in the manuscript:

- The characterization of the DTE molecules is insufficient. The authors should determine the photostationary state (PSS) of the material and further characterize the stability of the material. The authors claim: "that the compound has excellent reversible ring-open/ring-close photoisomerization behavior" (p. 5). The authors should produce data that shows at what dose the compound shows fatigue.

- In the system, together with the other building blocks and Eu as emitter, the authors have performed such fatigue tests and explain that the "SCP exhibited excellent fatigue resistance" – however, the system shows ~ 4% deterioration already after 20 switching cycles (p. 8). In my view this is not excellent especially with the sensitive application the authors have in mind. The authors should explain how this behavior could be improved or argue why this behavior were excellent – are there comparable but worse systems in use as anti-counterfeiting labels?

- The authors declare that the DTE molecule has a stable configuration. Correctly speaking, the molecule is bistable (open/close), as exchange between both states is kinetically hindered and therefore requires radiative energy for switching between the states. Thermodynamically, the two configurations will sit at two different energies making one more favorable than the other – but in real life the open and the closed form are stable. Otherwise, the materials would not be so interesting for the applications foreseen by the authors. The authors should rephrase this section on page 8.

- The authors declare that their material system is representative for "smart anticounterfeiting". I would like to know what is smart about the system especially compared with other approaches in use or in the literature.

- The authors need to describe better how the images in Figure 4 are produced – this means, how the QR-pattern is generated. It is not entirely clear, whether the luminescence is produced because the printed film was exposed through a mask that carried the QR-code pattern, or whether the printer was used to only print the ink in the here (figure 4b) luminescing regions.

- I understand that the pattern can be completely erased using UV light. However, when a good that is to be protected by the anticounterfeiting label is exposed to sunlight, then the pattern is visible. I do not understand in which way the printed QR code can prevent counterfeiting of goods. What is the mechanism behind anticounterfeiting. The authors declare that the "security pattern is invisible" (p. 10); however, exposure to sunlight will generate the fluorescent pattern – so it is not suitable as a secret anti-counterfeiting measure, or is it?

- The emission of the Eu-ions is at 620 nm and it overlaps with the absorption of the DTE that causes ring-opening (500 – 700 nm) and therefore less FRET and more Eu photoluminescence will result in a self-reinforcing or amplifying way. This entails that only the slightest amount of daylight or the slightest amount of luminescence from the Eu-ions will produce a luminescing pattern that grows stronger and stronger the longer it is illuminated. In my views, this self-switching is

detrimental for an application as an anticounterfeiting label.

Why did the authors not design a system, where the readout emission is situated in the spectrum between the two absorption bands (for opening and closing) of the DTE?

Response to Reviewer #1's Comments:

This is an interesting paper describing the preparation of photoswitchable supramolecular coordination polyelectrolyte as a smart anticounterfeiting ink. The authors regulate the FRET process between the lanthanide emitting center and the diarylethene component by light irradiation, and the resulting smart ink can be readily prepared in pure water and fully compatible with commercial printers. The features of environmentally friendly preparation process, remote light control, rapid response, and good fatigue resistance demonstrate a promising system as an anticounterfeiting ink in various fields. Thus, I recommend the publication of the work after addressing the following minor comments.

Reply: We thank the reviewer for the useful comments and recommendation.

(1) The authors assigned the absorption band at 294 nm in the UV-Vis spectrum of Eu^{3+} -L-OF-1 to the diarylethene unit. However, the absorption of 2,6-pyridinedicarboxylic acid is also near. Is there any overlap between their absorbance? In this context, UV-Vis spectrum of Eu^{3+} -L should be provided for comparison.

Reply: We thank the reviewer's useful advice. According to the comments, the UV-Vis spectrum of Eu^{3+} -L has been added to the revised Supplementary Information (Figure S18). As shown in Figure S18, 2,6-pyridinedicarboxylic acid shows a maximum absorption peak at 212 nm, and thus there is no obvious absorbance overlap between Eu^{3+} -L and OF-1.

(2) The authors measured the luminescence decays before and after UV light irradiation. I would suggest that they also measure and compare the luminescence quantum efficiency.

Reply: Thanks for the comments. According to the comments, the luminescence quantum yields of Eu^{3+} -L, and Eu^{3+} -L-OF-1 before and after UV light (300 nm) irradiation were measured to be 23.31%, 15.84% and 0.85%, respectively. The corresponding results have been added to the revised manuscript (page 5).

(3) The luminescence of the ink (Figure 3D inset) is not as bright as that in Figure S10, perhaps due to different lanthanide concentrations used. Nevertheless, the comparison of luminescence intensity between individual lanthanide complex and the ink should be provided.

Reply: We thank the reviewer's useful advice. The concentration of Eu^{3+} -L in Figure S10 is indeed higher than that in Figure 3D, thus showing higher brightness. To be noticed, the luminescence intensity of Eu^{3+} -L-OF-1 is lower than that of Eu^{3+} -L at the same concentration (Figure S32), perhaps due to the electron transfer between OF-1 and Eu^{3+} -L. However, the luminescence brightness of Eu^{3+} -L-OF-1 is still satisfied for anticounterfeiting patterning.

(4) In the elemental analysis, the authors only provide the C/H/N data of L and OF-1.

Elemental analysis results of major intermediates should also be given.

Reply: Thanks for the comments. Elemental analysis of major intermediates was carried out and the corresponding results have been added to the characterization data.

(5) The authors claimed that “the proton signals assigned to L became highly broadening after the coordination with Eu^{3+} ”, and indeed no obvious NMR peak was observed in Figure 1B. Have the authors tried to carry out this experiment at higher concentrations?

Reply: We thank the reviewer’s useful advice. We tried to monitor the NMR signal of Eu^{3+} -L at higher concentration, and the signal showing serious broadening. These results demonstrate that compound L and Eu^{3+} can form coordination polymer at relatively low concentration ($C_L = 2.1 \times 10^{-4}$ M, $C_{\text{Eu}} = 1.4 \times 10^{-4}$ M in Figure 1B).

Figure R1. ^1H NMR spectrum (DMSO: D_2O = 4:1, 400 MHz, 25 °C) of Eu^{3+} -L at high concentration ($C_L = 2.1 \times 10^{-3}$ M, $C_{\text{Eu}} = 1.4 \times 10^{-3}$ M).

Response to Reviewer #2's Comments:

Stimuli-responsive materials are important for smart anticounterfeiting. In this work, the authors report a photoresponsive anticounterfeiting ink composed of lanthanide complex and diarylethene unit in pure water. The mechanism behind was investigated and explained convincingly. The work is not only accessible, but also of interest and relevance to a broad readership. Compared to the reported stimuli-responsive materials dissolved/suspended in organic solvents, direct printing in pure water with commercial inkjet printer is especially appealing. The concept is full of attraction, and I believe that this is a breakthrough research in the field. Therefore, I recommend the publication of this interesting manuscript in Nature Communications after minor revisions as stated below:

Reply: We greatly appreciate the reviewer's useful comments and recommendation.

1. What is the major function of supramolecular coordination polyelectrolyte used? Could the same effect be easily accomplished by mixing the DAE molecule with Eu complex through simple solution processing? The relevant control experiments should be performed.

Reply: Thanks for the comments. In addition to the matched spectral overlap, FRET is also highly sensitive to the distance between donor and acceptor, and the donor and acceptor molecules must be in close proximity (1-10 nm, see: *J. Org. Chem.* 2010, 75, 3600; *Appl. Environ. Microbiol.* 2003, 69, 2330). In the reported diarylethene derivative based photoresponsive assembly systems, it also revealed that the distance between donor and acceptor is essential for the FRET (see: *J. Am. Chem. Soc.* 2013, 135, 10190). In our work, the supramolecular coordination polyelectrolyte makes the ink water-soluble, and the electrostatic interaction between Eu^{3+} -L and OF-1 in the supramolecular coordination polyelectrolyte can bring the distance between donor and acceptor closer. According to the comments, we have carried out some control experiments. As a model compound, Eu/2,6-pyridinedicarboxylic acid complex $\text{Eu}(\text{DPA})_3$ was mixed with compound **2** in EtOH. There was no electrostatic interaction between donor and acceptor when irradiated with 300 nm UV light for 60 seconds. Luminescence emission spectra revealed only 26% decrease as compared to individual $\text{Eu}(\text{DPA})_3$ at the same concentration, indicating no effective FRET occurred (Figure S33). This slight decrease may be attributed to accidental physical collision quenching. Our control experiments revealed that the supramolecular coordination polyelectrolyte is essential to draw the donor and acceptor in close proximity, which plays a vital role in facilitating high-efficiency FRET.

2. In addition to the Eu complex, can other lanthanide metal complexes or organic dyes with red emission be used for the same purpose?

Reply: Thanks for the comments. Theoretically, organic dyes having spectral overlap with DAE can be used to construct FRET systems in the present of driving force that draws the

donor and acceptor in close proximity. For example, FRET between DAE and porphyrin has been reported (see: *Angew. Chem. Int. Ed.* 2015, 54, 430). We have been engaging in the development of self-assembled lanthanide-containing luminescent hybrid materials. We here chose Eu complex because the anionic lanthanide coordination polymer is not only water soluble, but also could assemble with cationic DAE in pure water.

3. The absorption changes in the UV-Vis spectra upon UV and visible light irradiation, and the luminescence quenching curves showed a fast process, followed by a slow process. However, the luminescence recovery curve in Figure 3C showed a slow process in the early stage of visible light illumination. What reasons cause this phenomenon, and would this phenomenon affect the rapid response in smart anti-counterfeiting?

Reply: We thank the reviewer's careful examination. Actually, the luminescence recovery process is also rapid. However, the high energy laser (with excitation wavelength of 265 nm) of the luminescence analyzer leads to the isomerization of open-form diarylethene to its close-form conformation during the measurement. As shown in Figure S36, although the SCP solution changed back to colorless, the measuring point irradiated by the laser is still light blue, making the luminescence recovery process look slow.

4. In Figure 4, the authors first irradiated the pattern with 300 nm light, and then irradiated it with > 450 nm light. What if the pattern is directly irradiated with > 450 nm, and will it directly show luminescence just like when irradiated with UV light?

Reply: Thanks for the comments. The as-prepared **Eu³⁺-L-OF-1** exhibits characteristic luminescence of Eu³⁺ under UV light due to the "antenna effect" of DPA. This is because diarylethene was in its open form, and no photochromic FRET occurred. Directly irradiating **Eu³⁺-L-OF-1** pattern with >450 nm visible light would not cause any further color change under both daylight and 254 nm UV lamp.

5. For better clarity of the data, more specific explanations should be added to the caption of Figure 4. In Figure 4C and 4E, the background is dark blue, while in Figure 4D and 4F, it is light blue. Please make them consistent.

Reply: Thanks for the comments. To eliminate the ambiguity, we have added some discussions in the caption of Figure 4.

6. The FRET efficiency (E) can be calculated according to either the luminescence intensity or the luminescence lifetime data. Which method did the authors use? Is there any difference about the E values obtained from the two methods?

Reply: Thanks for the comments. In this manuscript, the E values calculated according to the luminescence intensity and lifetime were almost the same, and no obvious difference was found.

Response to Reviewer #3's Comments:

The paper by Zhao and Li et al reported a coordination supramolecular system displaying reversible photo-responsive emission and absorbance changes which is used for the purpose of anti-counterfeiting. The main problem of this work is lack of novelty since both the concept of anti-counterfeiting use photo-responsive materials and the compound **L** is quite old. Especially, **L** is well-studied 17~18 years ago(1. Vermondon T et al, *Macromolecules*, 2003, 36,7035; *Eur. J. Inorg. Chem.* 2003, 2847; *JACS*, 2004,126, 15802) and the coordination between **L** and Eu^{3+} was well-studied in a number of refs in recent years(Yang L et al, *Soft Matter*, 2011,7,2720;Xu et al, *Soft Matter*, 2014, 10, 4686; Wang et al, *Chem Commun*, 2013,49,3736; *Macromolecules* 2019, 52, 8643; *Soft Matter*, 2020, 16,2953.). It is a pity that Zhao et al is trying to hide this fact and never cite any of these references. In SI they cite some refs about the synthesis of the materials that is necessary for the synthesis of **L**, and deliberately citing ref 51 to support their conclusion about the coordination ratio between the DPA ligand. Actually, the coordination between **L** and Eu was already very clearly discussed in the above refs. The synthesis route of **L** they adopted is exactly the same as Vermondon et al. So, it is a pity that the authors never mention any of these work, and trying hard to pretend they are working on a new compound.

Reply: We appreciate the reviewer's critical comments. The reviewer's insightful suggestions have been thoroughly and carefully considered, and the corresponding revisions have been made.

About photoresponsive anticounterfeiting systems, there are indeed several reported photoresponsive anticounterfeiting examples (including but not limited to diarylethene derivative). However, they were mainly carried out in solid state (see: *Angew. Chem. Int. Ed.* 2019, 58, 18025; *J. Am. Chem. Soc.* 2017, 139, 16036), organic medium (see: *Nat. Commun.* 2018, 9, 3977; *Adv. Mater.* 2017, 29, 1605271) or in the form of gels (see: *Adv. Sci.* 2019, 6, 1901529; *Chem. Sci.* 2018, 9, 8019). Examples in pure water are very limited (see: *ACS Appl. Mater. Interfaces* 2018, 10, 39214, anti-counterfeiting ink was prepared in mixture of $\text{H}_2\text{O}/\text{DMF}$ in this reference, not in pure water). These pioneering studies gave us a lot of inspiration in designing the present work. While we conducted thorough literature search, we could only cite some relevant literature papers due to the page limitation. We did not try to hide the fact. We apologize for missing these crucial references, and thank you for pointing them out. These references have been added to the revised manuscript.

In terms of novelty for our work, 1) photoresponsive anticounterfeiting ink is readily prepared by mixing an anionic lanthanide coordination polymer with a cationic photochrome in pure water, and no complicated assembly and coating procedures is needed. 2) The ink does not contain organic solvents or toxic ions, enabling its usage in green conditions. 3) Our ink is fully compatible with commercial inkjet printer. The ink aqueous solution can be directly filled in cartridge to print various high-resolution QR codes, without the modification

of preexisting commercial inkjet printers. 4) Light irradiation offers clean trigger and spatiotemporal control over the anticounterfeiting patterns in a noninvasive manner.

Regarding compound **L**, we noticed that Vermonden T.; Cohen Stuart, M. A.; Wang, J.; Yan, Y. and their co-workers reported an interesting type of polyelectrolyte micelles based on a polycationic-neutral diblock copolymer and a polyanionic coordination polymer. The polyanionic coordination polymer is obtained by coordination between metal ions and a bis-ligand (L_2EO_4) containing two dipicolinic acid (DPA) moieties connected by a tetra-ethylene oxide spacer. Please note that this bis-ligand (L_2EO_4) is different to our compound **L**. In their work, the spacer between two DPA moieties in L_2EO_4 is tetra-ethylene oxide, while in our case, the spacer between two DPA moieties in compound **L** is C4 alkyl chain (Figure R2). According to the thorough literature search, Yin et al. firstly reported the synthesis of compound **L** (Ref S3, *Synth. Commun.* 2003, 33, 1113), and no further research about the coordination between compound **L** and metal ions (including transition metals and lanthanide ions) was reported since then. As the synthesis route we report here is different from this reference, the synthesis and characterization of compound **L** was stated in detail. The coordination behavior between **L** and Eu^{3+} was also investigated accordingly.

Figure R2. Chemical structures of L_2EO_4 reported by Vermonden T et al. and compound **L** in this work.

Other problems:

1. The coordination between **L** and Eu would not form polymer at concentrations they use. Only small coordination complex is possible since the coordination is reversible and dynamic. The experimental evidence for the formation of small complex can be found in refs by Vermonden, Yang, and Wang mentioned above;

Reply: Thanks for the comments. The concentrations for 1H NMR, luminescence emission and QR pattern printing experiments are $C_{Eu} = 1.4 \times 10^{-4}$ M, $C_L = C_{OF-1} = 2.1 \times 10^{-4}$ M. To confirm the formation of coordination polymer between **L** and Eu^{3+} at this concentration, we carried out dynamic light scattering (DLS) measurements. DLS experiment of Eu^{3+} -**L** (Figure 2A) shows hydrodynamic radius of 220 nm, providing intuitive evidence for the formation of large-scaled coordination polymer in solution. As the reviewer mentioned, the coordination is reversible and dynamic. Although the polymerization degree at this concentration may be relatively low, the DLS results and the signal broadening phenomenon in the 1H NMR spectra of Eu^{3+} -**L** jointly revealed the formation of the coordination polymer. Actually, we also tried higher concentrations, but the coagulation occurred (Figure R3) at the concentrations of C_{Eu}

= 2.8×10^{-4} M, $C_L = C_{OF-1} = 4.2 \times 10^{-4}$ M. Thus, the concentrations we used here are suitable for the presented studies. Please note that the concentrations for UV-Vis experiments are $C_{Eu} = 1.4 \times 10^{-5}$ M, $C_L = C_{OF-1} = 2.1 \times 10^{-5}$ M, because the UV absorbance of the system at 1H NMR concentration is too high.

Figure R3. Photographic image of Eu^{3+} -L-OF-1 at concentrations of $C_{Eu} = 2.8 \times 10^{-4}$ M, $C_L = C_{OF-1} = 4.2 \times 10^{-4}$ M.

2. They did not provide direct evidence for the formation of supramolecular assembly between Eu^{3+} -L and OF-1. Ionic interaction indeed occurs, but this does not mean the formation of supramolecular self-assembly. They should investigate the form of the ionic complexes.

Reply: Thanks for the comments. As shown in the DLS experiment, the hydrodynamic radius of Eu^{3+} -L-OF-1 increased to 500 nm (Figure 2A), much larger than that of Eu^{3+} -L, revealing that Eu^{3+} -L further assembled with OF-1 to form the supramolecular self-assembly, rather than individual small-scaled ionic complexes. The corresponding experiments and discussions have been added in the revised manuscript (pages 5-6).

3. Figure 1 and Figure S10, to study formation of Eu-L coordination complex, is not necessary since this coordination chemistry is already well-studied in literature.

Reply: Thanks for the comments. As discussed above, the coordination behavior between compound L and metal ions (including transition metals and lanthanide ions) has not been reported before. For better understanding the relationship between the coordination behavior and coordination polymer formation, it is reasonable to investigate the Eu^{3+} -L coordination complex in detail.

4. Using color change as anti-counterfeiting The QY for rare earth metal is usually below 21%. The authors need to discuss the QY for emission-based anti-counterfeiting.

Reply: We thank the reviewer's useful advice. According to the advice, the luminescence quantum yields of Eu^{3+} -L, and Eu^{3+} -L-OF-1 before and after UV light (300 nm) irradiation were measured to be 23.31%, 15.84% and 0.85%, respectively. The corresponding results have been added to the revised manuscript (page 5).

Response to Reviewer #4's Comments:

Z. Li, H. Li, and Y. Zhao report a material system based on bifunctional supramolecular building blocks that incorporate dithienylenthylene (DTE) photoswitches and Europium (Eu) ions to act as crosslinkers and emitters. In the assembled material, photoswitching of DTE results in a change in the electronic levels of the DTE making or preventing FRET from the Eu to the DTE. In the closed state, where FRET is possible, there is no emission from the Eu-ions and the material appears dark. In the open state there is no FRET and the material exhibits red photoluminescence typical for Eu-containing materials. The authors apply the molecular material as inks and produce QR codes using ink-jet printing. After deposition of the ink, the solvent evaporates, driving the material system into the above described supramolecular self-assembly. The authors foresee that a potential application of this material will be for anti-counterfeiting labels. In importance for such anti-counterfeiting systems is explained in the introduction of the paper.

The manuscript at hand represents a sensible collaboration of the two individual groups with their respective expertise in supramolecular assembly and photonic material systems. While I find the work suitable for nature communications, I have the following questions and remarks, which are not clearly described in the manuscript:

Reply: We gratefully appreciate the reviewer's encouraging and insightful comments that significantly improve the scientific accuracy of our manuscript.

1. The characterization of the DTE molecules is insufficient. The authors should determine the photostationary state (PSS) of the material and further characterize the stability of the material. The authors claim: "that the compound has excellent reversible ring-open/ring-close photoisomerization behavior" (p. 5). The authors should produce data that shows at what dose the compound shows fatigue.

Reply: Thanks for the comments. According to the comments, we determined the photostationary state of individual DTE (compound **1**) by UV-Vis spectra (Figure S19-S22) and ¹H NMR spectra (Figure S23 and S24). As shown in Figure S19, the UV-Vis spectrum of OF-**1** showed a strong absorption at 294 nm ($\epsilon = 3.9 \times 10^4 \text{ cm}^{-1} \text{ M}^{-1}$), and no absorption >400 nm was observed. Upon the irradiation with 300 nm UV light, the absorption band at 294 nm decreased gradually and new peaks appeared at 380 and 596 nm ($\epsilon = 7.8 \times 10^3 \text{ cm}^{-1} \text{ M}^{-1}$) with an isosbestic point at 323 nm. These changes reached photostationary state in 60 s (Figure S20), leading to an obvious color change of the solution from colorless to blue (Figure S19A, inset). A combination of these phenomena indicates the transition from OF-**1** to CF-**1**. Interestingly, a complete recovery to the original UV-Vis spectrum of OF-**1** was achieved upon subsequent irradiation of CF-**1** with visible light ($\lambda > 450 \text{ nm}$) in 60 s, accompanied with the color change of blue solution back to colorless (Figure S19B and S21). Notably, this ring-open/ring-close photoisomerization cycle was repeatable for at least 20 times without

apparent fatigue (Figure S22), suggesting the good reciprocation of this process.

To further clarify the isomerization behavior of compound **1**, ^1H NMR spectra of OF-**1** in a mixed solvent were recorded (Figure S23). Upon 300 nm light irradiation, the methyl protons (H_a) showed a downfield shift from 1.88 ppm to 2.00 ppm, the thiophene protons (H_b) underwent a drastic upfield shift from 7.30 ppm to 6.81 ppm, and the aromatic protons (H_c and H_d) presented downfield shifts ($\Delta\delta = 0.11$ ppm for H_c and 0.06 ppm for H_d). The molar ratio of CF-**1** : OF-**1** was determined to be 0.95:0.05 according to the integrating resonance of proton H_a (Figure S24), indicating nearly quantitative transition from OF-**1** to CF-**1** after 300 nm light irradiation. After subsequent irradiation with visible light, the CF-**1** isomer could be converted back to OF-**1**, as shown by the recovery of the original ^1H NMR spectrum (Figure S23C). The corresponding experiments and discussions have been added in the Supplementary Information (pages S12 and S14).

2. In the system, together with the other building blocks and Eu as emitter, the authors have performed such fatigue tests and explain that the “SCP exhibited excellent fatigue resistance” – however, the system shows $\sim 4\%$ deterioration already after 20 switching cycles (p. 8). In my view this is not excellent especially with the sensitive application the authors have in mind. The authors should explain how this behavior could be improved or argue why this behavior were excellent – are there comparable but worse systems in use as anti-counterfeiting labels?

Reply: We thank the reviewer’s insightful comments. This deterioration maybe explained as follow: the fatigue resistance of the SCP was determined by luminescence spectra, and as compared to the luminescence quenching process, the luminescence recovery process may seem slow (Figure 3B and 3C). Actually, the luminescence recovery process is also rapid. However, the high energy laser (with excitation wavelength of 265 nm) of the luminescence analyzer leads to the isomerization of open-form diarylethene to its close-form conformation during the measurement. As shown in Figure S36, although the SCP solution recovered back to colorless, the measuring point irradiated by the laser is still light blue, causing the measured value lower than the true value. Thus, this $\sim 4\%$ deterioration may come from the measurement error, since the light-driven photoisomerization cycle of individual compound **1** can be repeatable for at least 20 times without any fatigue, showing excellent fatigue resistance (determined by UV-Vis spectra shown in Figure S22). On the other hand, $\sim 4\%$ deterioration after 20 consecutive cycles is acceptable, as the remote light triggered information pattern with visible/invisible transformation process still remained unaffected.

3. The authors declare that the DTE molecule has a stable configuration. Correctly speaking, the molecule is bistable (open/close), as exchange between both states is kinetically hindered and therefore requires radiative energy for switching between the states. Thermodynamically, the two configurations will sit at two different energies making one

more favorable than the other – but in real life the open and the closed form are stable. Otherwise, the materials would not be so interesting for the applications foreseen by the authors. The authors should rephrase this section on page

Reply: We thank the reviewer's useful comments. To eliminate the ambiguity, we have removed the inaccurate statements and rephrased this section in the revised manuscript (page 9).

4. The authors declare that their material system is representative for “smart anticounterfeiting”. I would like to know what is smart about the system especially compared with other approaches in use or in the literature.

Reply: We thank the reviewer's comments. Materials with static luminescent outputs are adverse to their practical applications in anticounterfeiting, because the information recorded directly in these materials is usually visible under either ambient conditions or the excitation of NIR or UV light. In this context, smart luminescent materials, that can change their luminescent outputs in response to external stimuli, have been regarded as suitable candidates to prevent counterfeiting. However, some of these responses are irreversible. As a result, the quenched luminescence cannot be recovered anymore. The encoded information is irreversibly destroyed, rather than temporarily hidden. Thus, the ones with luminescence on-off switch behavior are extremely interesting, as reversible visible/invisible information transformation can be achieved. On the other hand, invasive stimuli-responsive luminescent materials often rely on constant addition of chemicals or heat, which may contaminate or destroy the goods, thus limiting their applications in authenticating food and medicine. In contrast, photoresponsive luminescent materials are appealing candidates, because light irradiation has the possibility to achieve a remote trigger and have clean and spatiotemporal control over the action with high precision.

In the present SCP, the ring-close and ring-open photoisomerization of the diarylethene moiety regulates the FRET process, leading to reversible luminescence on/off switch in SCP capable of multiple information authentication. Light irradiation offers clear trigger and spatiotemporal control over the anticounterfeiting patterns in a noninvasive manner. More importantly, water is the only solvent used in preparing the security ink, enabling its usage in a green condition and its good compatibility with commercial printers.

5. The authors need to describe better how the images in Figure 4 are produced – this means, how the QR-pattern is generated. It is not entirely clear, whether the luminescence is produced because the printed film was exposed through a mask that carried the QR-code pattern, or whether the printer was used to only print the ink in the here (figure 4b) luminescing regions.

Reply: We thank the reviewer's comments. The PET film was firstly printed with blue background, and the QR-pattern was then directly printed on the blue PET film. The detailed

preparation procedure has been added to the experimental section of the revised manuscript.

6. I understand that the pattern can be completely erased using UV light. However, when a good that is to be protected by the anticounterfeiting label is exposed to sunlight, then the pattern is visible. I do not understand in which way the printed QR code can prevent counterfeiting of goods. What is the mechanism behind anticounterfeiting. The authors declare that the “security pattern is invisible” (p. 10); however, exposure to sunlight will generate the fluorescent pattern – so it is not suitable as a secret anti-counterfeiting measure, or is it?

Reply: We thank the reviewer’s comments. As discussed above, the QR-pattern was printed on the blue PET film. The obtained QR code was invisible under daylight due to the colorless nature of Eu^{3+} -L-OF-1 aqueous solution. However, bright red luminescent pattern was observed under 254 nm UV lamp, allowing to retrieve the encoded information. Upon 300 nm UV light irradiation, Eu^{3+} -L-OF-1 transformed to Eu^{3+} -L-CF-1, and the luminescence was quenched. Meanwhile, the blue color of Eu^{3+} -L-CF-1 was also masked by the blue background of the PET film, thus making the QR pattern invisible under both daylight and UV light. Please note that the exposure to sunlight would not result in the recovery of the luminescent pattern, because only visible light with specific wavelength (>450 nm) can lead to the photoisomerization of CF-1 back to OF-1, rather than sunlight.

7. The emission of the Eu-ions is at 620 nm and it overlaps with the absorption of the DTE that causes ring-opening (500 – 700 nm) and therefore less FRET and more Eu photoluminescence will result in a self-reinforcing or amplifying way. This entails that only the slightest amount of daylight or the slightest amount of luminescence from the Eu-ions will produce a luminescing pattern that grows stronger and stronger the longer it is illuminated. In my views, this self-switching is detrimental for an application as an anticounterfeiting label. Why did the authors not design a system, where the readout emission is situated in the spectrum between the two absorption bands (for opening and closing) of the DTE?

Reply: We thank the reviewer’s comments. Spectral overlap of the donor emission and the acceptor absorption is the prerequisite for FRET, so that the donor emission can be absorbed by the acceptor. Thus, to design DTE-based photoresponsive FRET systems, we have to choose emitting centers with emission wavelength overlapping with the close-form DTE (red emission in the range of 500-700 nm), since the open-form DTE only shows absorption in the UV region. Emission (for example 450 nm) situated in the spectrum between the two absorption bands would not generate FRET behavior. We here chose Eu complex with emitting center at 615 nm as the donor. Photoresponsive FRET systems based on red emitting dyes (such as porphyrin derivatives with emission center at 650 nm) and DTE were

also reported (see: *Angew. Chem. Int. Ed.* 2015, 54, 430; *Adv. Optical Mater.* 2017, 1700770). No self-reinforcing or amplifying was observed in all of these references and our work, probably because the photoluminescence of donor is too weak to drive the photoisomerization of DTE molecule.

Reviewers' Comments:

Reviewer #1:

Remarks to the Author:

In this revised version, the authors fully addressed the concerns of the reviewers, thus should be suitable for publication.

Reviewer #2:

Remarks to the Author:

Publish as it is.

Reviewer #3:

Remarks to the Author:

The authors have replied my concerns about the references. I am sorry for not noticing that the spacer in the ligand they use is different from that used in literature. But I do think the coordination between DPA and Eu is the same whatever spacer chain is used. It is not necessary to cite all of those references; I just want to remind the authors to cite proper refs.

The authors use DLS measurements to support the formation of self-assembled structures, this is rather misleading. DLS only detects the presence of colloidal sized objects and fit the diffusion constant using model specified for spheres. Any irregular clusters can be detected in DLS and be assigned a size by DLS. The authors are suggested to provide direct evidence of the formation of self-assembly using TEM. Otherwise, the possibility of forming ionic clusters cannot be ruled out. Actually, since the FRET system is composed of oppositely charged species, it is very likely to form just clusters rather than self-assembled structures where the component molecules arrange orderly, which is the meaning of self-assembly.

Reviewer #4:

Remarks to the Author:

The authors have addressed all questions by the referees; however, in my view the authors do not provide sufficient evidence to confirm that the material system is not prone to severe self-switching.

In fact I am not the only reviewer that picked up on this problem. While the authors claim that the FRET process is an integral part of their material system, it actually turns out as a bug, which leads to self-switching in the presence of sun light.

The authors claim that only light "> 450 nm" (which includes the visible spectrum and therefore also sunlight or ambient lighting) can switch the DTE molecule, and the switching requires sufficient dose - so it requires a laser and not a sun or ambient lightsource.

However, this evidence is not provided. In my view also sunlight should switch the molecule, but maybe not at a sufficiently high rate - however, since the self-switching process would be self-amplifying I suspect that at a certain dose the process will increase in rate even at ambient lighting.

Also the authors claim that it depends on the type of experiment whether they observe fatigue or not. The authors should explain in the manuscript why that is and provide numbers/data to the fatigue problem - an estimation of the half-life on the material system would probably show how superior their system indeed is.

I cannot recommend acceptance of this manuscript, before the authors do not provide the additional data several referees have requested already in round one of the reviewing process.

List of Changes

- (1) Pages 6: according to reviewers' comments, TEM images and related discussions are added.
- (2) Pages 9: according to reviewers' comments, the half-life ($t_{1/2}$) results of **Eu³⁺-L-CF-1** and related discussions are added.

Reviewer #1 (Remarks to the Author):

In this revised version, the authors fully addressed the concerns of the reviewers, thus should be suitable for publication.

Reply: We greatly appreciate the reviewer's positive recommendation of publication.

Reviewer #2 (Remarks to the Author):

Publish as it is.

Reply: We greatly appreciate the reviewer's positive recommendation of publication.

Reviewer #3 (Remarks to the Author):

The authors have replied my concerns about the references. I am sorry for not noticing that the spacer in the ligand they use is different from that used in literature. But I do think the coordination between DPA and Eu is the same whatever spacer chain is used. It is not necessary to cite all of those references; I just want to remind the authors to cite proper refs. The authors use DLS measurements to support the formation of self-assembled structures, this is rather misleading. DLS only detects the presence of colloidal sized objects and fit the diffusion constant using model specified for spheres. Any irregular clusters can be detected in DLS and be assigned a size by DLS. The authors are suggested to provide direct evidence of the formation of self-assembly using TEM. Otherwise, the possibility of forming ionic clusters cannot be ruled out. Actually, since the FRET system is composed of oppositely charged species, it is very likely to form just clusters rather than self-assembled structures where the component molecules arrange orderly, which is the meaning of self-assembly.

Reply: We thank the reviewer's constructive comments. According to the comments, we have carried out TEM experiments to further characterize the intuitive morphology of $\text{Eu}^{3+}\text{-L-OF-1}$. Uniform spheres with average diameter of 300 nm were observed from the TEM images (Figure S26), providing useful evidence for the formation of self-assembly between $\text{Eu}^{3+}\text{-L}$ and OF-1. In addition to the electrostatic interaction between adjacent $\text{Eu}^{3+}\text{-L}$ and OF-1, electrostatic interaction would also exist in a long-range manner between the oppositely charged species, which may be responsible for the dense sphere morphology. The corresponding experiments and discussions have been added in the revised manuscript (page 6).

Reviewer #4 (Remarks to the Author):

The authors have addressed all questions by the referees; however, in my view the authors do not provide sufficient evidence to confirm that the material system is not prone to severe self-switching. In fact I am not the only reviewer that picked up on this problem. While the authors claim that the FRET process is an integral part of their material system, it actually turns out as a bug, which leads to self-switching in the presence of sun light. The authors claim that only light "> 450 nm" (which includes the visible spectrum and therefore also sunlight or ambient lighting) can switch the DTE molecule, and the switching requires sufficient dose - so it requires a laser and not a sun or ambient lightsource. However, this evidence is not provided. In my view also sunlight should switch the molecule, but maybe not at a sufficiently high rate - however, since the self-switching process would be self-amplifying I suspect that at a certain dose the process will increase in rate even at ambient lighting. Also the authors claim that it depends on the type of experiment whether they observe fatigue or not. The authors should explain in the manuscript why that is and provide numbers/data to the fatigue problem - an estimation of the half-life on the material system would probably show how superior their system indeed is. I cannot recommend acceptance of this manuscript, before the authors do not provide the additional data several referees have requested already in round one of the reviewing process.

Reply: We appreciate the reviewer's critical comments. According to the comments, we have calculated the half-life and activation energy of **Eu³⁺-L-CF-1** according to the reported method (see: *J. Am. Chem. Soc.* **2000**, *122*, 4871; *J. Mater. Chem. C* **2019**, *7*, 2865; *J. Phys. Chem. C* **2019**, *123*, 31212). The half-life ($t_{1/2}$) of **Eu³⁺-L-CF-1** at 25 °C was estimated to be 376.7 min (Figure S38), ranking one of the longest $t_{1/2}$ values reported so far in diarylethene derivatives. The activation energy was calculated to be 13 kJ mol⁻¹ according to the first-order rate constants of **Eu³⁺-L-CF-1** at different temperatures (Table S1 and Figure S39).

On one hand, sunlight contains both UV and visible spectra, and thus the self-switching is ultraslow. On the other hand, the half-life of **Eu³⁺-L-CF-1** is very long, and no self-switching amplification or acceleration was observed upon continuous exposure to sunlight (Figure S38), probably because that the energy of sunlight dose not reach the activation energy of **Eu³⁺-L-CF-1**. As shown in Figure S40, only slight self-switching was observed after continuous exposure of **Eu³⁺-L-CF-1** to sunlight for 90 min, accompanied by negligible color change (Figure S40 inset). To be noticed, solid **Eu³⁺-L-CF-1** is much more stable than that in solution, as confirmed by that the erased pattern remained unreadable even after placing it under sunlight for one month (Supplementary Movies S5 and S6, and related discussions in page 11). Thus, the utilization of our system for printable smart anticounterfeiting is tenable.

About the fatigue problem, the fatigue resistance of our system is comparable to or even better than the reported diarylethene derivatives (e.g., *J. Am. Chem. Soc.* **2013**, *135*, 10190; *Inorg. Chem.* **2016**, *55*, 7962; *Chem. Eur. J.* **2017**, *23*, 14425). According to advice, the half-life

($t_{1/2}$) of our system was determined (as shown above) to show the good stability. In addition, the fatigue performance in the present work can meet the application requirements for multiple anticounterfeiting, since the remote light triggered information patterns with visible/invisible transformation process still remain unaffected after 20 consecutive cycles based on the fatigue resistance data (Figure 3D).

Overall, as a proof of concept, this work presents a viable alternative for photoresponsive smart anticounterfeiting. Future improvements such as developing more stable diarylethene derivatives with longer half-life are still a promising research direction.

Corresponding experiment results and discussions have been added in the revised manuscript (page 9).

Reviewers' Comments:

Reviewer #3:

Remarks to the Author:

I am satisfied the revision. Published as it is.

Reviewer #4:

Remarks to the Author:

I am happy with the additional evidence that the authors provided. The manuscript should now be published.

Reviewer #3 (Remarks to the Author):

I am satisfied the revision. Published as it is.

Reply: We greatly appreciate the reviewer's recommendation of publication.

Reviewer #4 (Remarks to the Author):

I am happy with the additional evidence that the authors provided. The manuscript should now be published.

Reply: We greatly appreciate the reviewer's recommendation of publication.